# An Optimized Protocol for Enzymatic Hypothiocyanous Acid Synthesis

**DOI:** 10.3390/mps8060144

**Published:** 2025-12-01

**Authors:** Alexander I. Kostyuk, Gleb S. Oleinik, Vladimir A. Mitkevich, Vsevolod V. Belousov, Alexey V. Sokolov, Dmitry S. Bilan

**Affiliations:** 1Institute of Translational Medicine, Pirogov Russian National Research Medical University, 117997 Moscow, Russia; alexander.kostyuk@inbox.ru (A.I.K.); belousov@fccps.ru (V.V.B.); 2M.M. Shemyakin and Yu.A. Ovchinnikov Institute of Bioorganic Chemistry, Russian Academy of Sciences, 117997 Moscow, Russia; gleboleynik.2003@mail.ru; 3Faculty of Biology, M.V. Lomonosov Moscow State University, 119234 Moscow, Russia; 4Engelhardt Institute of Molecular Biology, Russian Academy of Sciences, 119991 Moscow, Russia; mitkevich@gmail.com; 5Federal Center of Brain Research and Neurotechnologies, Federal Medical Biological Agency, 117513 Moscow, Russia; 6Life Improvement by Future Technologies Center, 121205 Moscow, Russia; 7Department of Molecular Biology of Viruses, Smorodintsev Research Institute of Influenza, Ministry of Health of the Russian Federation, 197022 Saint-Petersburg, Russia

**Keywords:** enzymatic synthesis, HOSCN, hypohalous acids, hypothiocyanous acid, lactoperoxidase

## Abstract

Investigation of molecular mechanisms that underlie the toxicity of reactive oxidants requires the usage of reductionist cellular models, where laboratory cultures are treated by known doses of the target compounds in strictly controlled conditions. In recent years, much attention has been focused on hypothiocyanous acid (HOSCN), a pseudohypohalous acid that is one of the main products of chordata heme peroxidases. Due to its instability, HOSCN cannot be purchased and stored, so it has to be enzymatically synthesized before each experiment. For the first time, we systematically classified the published protocols for HOSCN synthesis, compared them by product yield, and found that the highest achievable concentration was about 1.9 mM. This value is not convenient for large-scale experiments with high cell density. Therefore, we developed an improved protocol for HOSCN preparation by optimizing reagent ratios, incubation times, and temperature. The current paper describes all steps from scratch, namely lactoperoxidase purification via a combination of cation exchange, hydrophobic interaction, and size exclusion chromatography, HOSCN synthesis from SCN^−^ and H_2_O_2_, as well as HOSCN concentration measurement. The main advantage of the current protocol is that the product yield reaches 2.9 mM, which is 60% higher than published alternatives.

## 1. Introduction

Chordata heme peroxidases are one of the first lines of defense against pathogenic microorganisms that enter our bodies. They perform this function by converting H_2_O_2_, which is produced by NADPH oxidases, into more kinetically active compounds, (pseudo)hypohalous acids (HOCl, HOBr, and HOSCN) [1]. For instance, reactions between H_2_O_2_ and free thiolates are characterized by second-order rate constants of about 20 M^−1^ s^−1^ [2], while similar processes with HOCl have values of ~10^8^ M^−1^ s^−1^ [3]. This inevitably affects their ability to migrate through tissues and the spectrum of biologically relevant targets. However, (pseudo)hypohalous acids are very different themselves. Thus, in contrast to HOCl and HOBr, which are capable of modifying almost all functional groups (including amines, unsaturated bonds, or aromatic moieties), HOSCN reactivity is limited to thiols and selenols [4]. It has been suggested that this selectivity is important for protecting the host tissues from halogenating stress, because thioredoxin reductase demonstrates a certain ability to detoxify this agent, while bacterial homologues lack this function [5]. However, in recent years, reports have been emerging that similar systems have also evolved in procaryotes [6,7].

In living organisms, (pseudo)hypohalous acids are produced in a mixture due to the broad substrate specificity of corresponding enzymes [8], which is further complicated by the fact that blood concentrations of SCN^−^ can vary significantly depending on lifestyle [9]. Considering physiological (pseudo)halide ion levels, myeloperoxidase mostly generates HOCl and HOSCN, eosinophil peroxidase—HOBr and HOSCN—and lactoperoxidase—HOSCN only [1]. At the same time, various external factors can affect the catalytic properties of heme peroxidases. Thus, the alkalization of the medium, which can happen in the lumen of neutrophil phagosome, increases HOBr production by myeloperoxidase [10]. However, in most situations, HOCl and HOBr are not believed to be the primary agents of halogenating stress. They are too reactive to migrate long distances, namely, it is expected that the radius of HOCl action is less than 0.1 μm [11], which is ~0.05 of *Escherichia coli* length. Apparently, they transfer their halogen moieties to amines that are present in high concentrations in all biological fluids and the resulting halamines are responsible for the expansion of the oxidative stress region. Finally, it should be mentioned that different halamines demonstrate dramatically different reactivities and, therefore, have different subsets of targets [12]. SCN^−^ mainly originates from cyanide neutralization by rhodanese, so its relatively high consumption either directly (like smoking) or indirectly (in the form of cyanogenic glycosides, e.g., from cassava) inevitably shifts the profile of halogenating agents [9]. Mathematical modeling suggests that, in the case of smokers, more than one-third of H_2_O_2_, consumed by myeloperoxidase, is converted to HOSCN, not to HOCl [13]. Moreover, HOCl and HOBr can nonenzymatically react with SCN^−^, leading to further interconversion of the oxidants [14].

At this moment, strong evidence exists that out-of-control halogenating stress underlies many socially significant diseases. Historically, most attention was focused on cardiovascular pathologies, like atherosclerosis and coronary syndromes, where (pseudo)hypohalous acids induce matrix remodeling, endothelial dysfunction, and smooth muscle cells death [15]; however, more and more data indicate their role in neurodegenerative conditions [16] and cancer [17]. Unfortunately, we still do not fully understand the exact molecular mechanisms that are at the core of (pseudo)hypohalous acids’ toxicity. HOCl has been widely studied in this context, but the obtained results are often contradictory. Much less is known about the mechanisms by which HOSCN damages cells. Experimental evidence exists that it leads to the formation of sulfenic acids in matrix proteins, inhibits the respiratory chain, and triggers mitochondrial permeability transition pore-mediated potential reduction with subsequent release of cytochrome c, apoptosis inducible factor, and endonuclease G into the cytosol [18,19,20]. However, it is generally accepted that SCN^−^ has a protective function both in vitro and in vivo [19,21,22,23,24]. As its concentration increases, more oxidative equivalents are channeled to HOSCN rather than HOCl, and the metabolic changes it causes are characterized by greater reversibility [24]. Nevertheless, in some models, such as murine macrophages, HOSCN appears to be even more toxic than HOCl [18,25].

At inflammatory sites in vivo, tissues encounter a complex mixture of oxidants and signaling molecules that determine their fate. Unraveling the tangle of these interactions is impossible without reductionist cellular models, where laboratory cultures are treated by known doses of the target compounds in strictly controlled conditions. This is especially important in the case of (pseudo)hypohalous acids, since they are reactive enough to be consumed by the components of the medium that contain sulfur or nitrogen atoms. While HOCl, HOBr, and chloramines (e.g., N-chlorotaurine) can be easily purchased or synthesized and stored for months, HOSCN is extremely unstable and has to be prepared before each experiment [26]. It may seem that mixing HOCl with SCN^−^ is the easiest way to synthesize HOSCN in laboratory conditions. However, it should be kept in mind that HOSCN is also capable of reacting with HOCl, which leads to overoxidation [26]. It is, therefore, required to use huge excess of SCN^−^ and alkaline medium to obtain a relatively pure product [27]. However, achieving a stoichiometric yield is a hard task in this case as well.

Since it is difficult to control for the physiological effects of overoxidized species, enzymatic HOSCN synthesis with lactoperoxidase (LPO) remains the most popular technique. Multiple articles include corresponding protocols, which usually differ by several factors like SCN^−^/H_2_O_2_ ratio, buffer pH value, the number of substrate additions, and incubation time. In practice, it is almost impossible to rationally choose one of them, because the authors usually do not report the mean yields. Only several papers provide rough estimations. The complex nature of the kinetic cycles of LPO, as well as the remarkably low stability of HOSCN in aqueous solutions, are the main factors that complicate the rational development of synthesis strategies. For instance, it was experimentally shown that, when present at a high concentration (relative to SCN^−^), excess H_2_O_2_ can tightly bind to the enzyme and disrupt its work [28]. Moreover, it converts LPO into Compound III with subsequent heme cleavage and iron liberation, which leads to irreversible inactivation [29]. At the same time, structural data suggest that SCN^−^ can also act as an LPO inhibitor by binding at a distal site and, therefore, restricting the accessibility to H_2_O_2_ [30]. Another molecule capable of reorganizing the microenvironment of the heme and consequently affecting catalysis is HOSCN itself [31]. Finally, cyanide is one of the products that may emerge during HOSCN decomposition [32], and it is a well-known inhibitor of LPO [33].

Some experiments require relatively large concentrations of HOSCN. In our practice, for certain lines, 2.5 pmol of HOSCN per cell is a sublethal dose, so for an annexin V test with a flow cytometry readout, we have to use 1 mL of 2.5 mM oxidant in these conditions (per 1 million cells, which is a requirement from the kit’s manufacturer). In our laboratory, we found that by following the published guidelines, it is not possible to obtain a HOSCN stock with a concentration of more than 1.86 mM (for additional information, see Section 4 Expected Results).

Moreover, large-scale in vitro studies may require relatively high quantities of LPO. Therefore, it may be more cost-effective to purify the enzyme from scratch rather than purchase a commercial reagent. LPO is a member of the chordata heme peroxidases family [34], which requires the presence of SCN^−^ and hydrogen peroxide to exert antimicrobial effects in exocrine secretions [35]. In contrast to human breast milk, which contains only traces of LPO [36], in bovine milk, LPO is the most abundant enzyme after xanthine oxidase and, in this case, its concentration is around 30 mg per L [37]. For isolation of LPO (pI about 9.6, molecular weight about 77.5 kDa) from bovine milk, the process of purification usually starts with cation-exchange chromatography (e.g., sorbents with sulphopropyl or carboxymethyl groups), and fractions eluted by a high concentration of salt are subjected to hydrophobic resins (e.g., sorbents with phenyl or butyl groups) [38]. The second procedure is aimed to separate LPO from lactoferrin, which is characterized by a close pI value and almost the same molecular weight. Considering high content of casein in bovine milk and dramatic decrease in LPO activity after pasteurization, unpasteurized bovine milk treated by citric acid for precipitation of casein is a suitable source material for purification. In the final steps of LPO isolation, its purity can be determined as the ratio of absorbance at 412 nm (Soret band of heme) and at 280 nm (aromatic amino acid residues of the protein). This value, named Reinheitszahl (RZ), should be approximately 0.95 for pure bovine LPO solution. The optical spectrum of LPO Fe(III) demonstrates a Soret band at 412 nm with an extinction coefficient of 112.3 mM^−1^ cm^−1^, which can be used for measuring concentration [37].

The current paper, to our knowledge, is the first published attempt to systematically gather information on LPO purification, HOSCN enzymatic synthesis, and concentration measurement to a single step-by-step protocol that can be easily reproduced in most laboratories. Its biggest advantage, besides clarity, high granularity, and coverage of all HOSCN synthesis aspects, is the high product yield. Thus, by following all the steps, it is possible to prepare 2.9 mM oxidant solution, which is a 60% yield improvement compared to the best recipes from the literature (Bozonet et al. [39] and van Leeuwen et al. [40]; the resulting HOSCN concentration is about 1.9 mM). This protocol, therefore, is suitable for large-scale in vitro experiments, which include redox stress modeling in big cell populations.

## 2. Experimental Design

The protocol described here consists of the following two main parts: LPO purification and HOSCN synthesis itself. LPO purification starts with casein precipitation by citric acid and is followed by the following three chromatography steps: (1) cation exchange chromatography (allows for capturing LPO); (2) hydrophobic interaction chromatography (allows for separating LPO from lactoferrin); and (3) size exclusion chromatography (allows for eliminating the residual impurities). Then, the quality of the purified enzyme can be accessed via ABTS assay (measures LPO activity) and Reinheitszahl reading (indicates of the enzyme purity). HOSCN synthesis is performed in Eppendorf-like tubes with H_2_O_2_ and NaSCN as the source materials, and the yield can be measured in TNB assay. The general scheme of the protocol can be found in Figure 1.

### 2.1. Materials

Unpasteurized bovine milk collected and stored at +4–8 °C for no more than one day before freezing at −18 °C (stable at −18 °C for at least 6 months).Ammonium Sulfate ((NH_4_)_2_SO_4_) (Sigma-Aldrich, 7783-20-2, St. Louis, MO, USA).Bovine Liver Catalase (Sigma-Aldrich, 9001-05-2, St. Louis, MO, USA).Capsule membrane filter, pore size of 0.45 µm, polyether sulphone (Sterlitech, 1470233, Auburn, WA, USA).Citric Acid Monohydrate (C_6_H_8_O_7_·H_2_O) (Sigma-Aldrich, 5949-29-1, St. Louis, MO, USA).Clear, uncoated, flat-bottom 96-well plates, preferably divided into 8-well strips (Corning, 96-well Clear Polystyrene Not Treated Stripwell™ Microplate, 2593, Corning, NY, USA).Guaiacol ((CH_3_O)C_6_H_4_OH) (Sigma-Aldrich, 90-05-1, St. Louis, MO, USA).Hydrogen Peroxide, 8.5 M (H_2_O_2_) (Sigma-Aldrich, 7722-84-1, St. Louis, MO, USA).Hydrophobic resin for protein separation packed in a chromatography column (Cytiva, Butyl-Sepharose Fast Flow, 17098001, Marlborough, MA, USA).Size-exclusion resin suitable for 78 kDa-protein separation packed in chromatographic column (Cytiva, Superdex 200, 17104301, Marlborough, MA, USA).Sodium Chloride (NaCl) (Sigma-Aldrich, 7647-14-5, St. Louis, MO, USA).Sodium Hydroxide (NaOH) (Sigma-Aldrich, 1310-73-2, St. Louis, MO, USA).Sodium Phosphate Dibasic Heptahydrate (Na_2_HPO_4_·7H_2_O) (Sigma-Aldrich, 7782-85-6, St. Louis, MO, USA).Sodium Phosphate Monobasic Monohydrate (NaH_2_PO_4_·H_2_O) (Sigma-Aldrich, 10049-21-5, St. Louis, MO, USA).Sodium Thiocyanate (NaSCN) (Sigma-Aldrich, 540-72-7, St. Louis, MO, USA).Strong cation exchanger resin for protein separation packed in a chromatography column (Bio-Rad, UNOSphere S, 156-0113, Hercules, CA, USA).Ultrafiltration units, MWCO 10 kDa (Sartorius, Vivaspin 500 columns, VS0101, Göttingen, Germany).Ultrafiltration units, MWCO 30 kDa (Sartorius, Vivaspin 20 columns, VS2022, Göttingen, Germany).UV transparent cuvettes (Sarsted, 67.759, Nümbrecht, Germany).2,2′-Azinobis-(3-Ethylbenzothiazoline-6-Sulphonic) Diammonium Salt (ABTS) (Sidma-Aldrich, 30931-67-0, St. Louis, MO, USA).15 mL conical tubes (SPL Life Sciences, 50115, Pocheon-si, Republic of Korea).50 mL conical tubes (SPL Life Sciences, 50150, Pocheon-si, Republic of Korea).0.5–10 μL disposable tips (ULPlast, OM-10-RF-C, Warszawa, Poland).1–200 μL disposable tips (ULPlast, OM-200-RF-Y, Warszawa, Poland).100–1000 μL disposable tips (ULPlast, OM-1000-RF-B, Warszawa, Poland).5,5-Dithio-bis-(2-Nitrobenzoic Acid) (DTNB) (Sigma-Aldrich, 69-78-3, St. Louis, MO, USA).0.5 mL Eppendorf-type tubes (SSIbio, 1310-00, Lodi, CA, USA).2 mL Eppendorf-type tubes (SSIbio, 1110-00, Lodi, CA, USA).2-Morpholinoethanesulfonic Acid Monohydrate (MES·H_2_O) (Sigma-Aldrich, 145224-94-8, St. Louis, MO, USA).

### 2.2. Equipment

Automatic pipette, 0.5–10 μL (Eppendorf, 3123000020, Hamburg, Germany).Automatic pipette, 10–100 μL (Eppendorf, 3123000047, Hamburg, Germany).Automatic pipette, 100–1000 μL (Eppendorf, 3123000063, Hamburg, Germany).Centrifuge capable of generating 2500× *g* acceleration and cooling samples (Eppendorf, 5810R, Hamburg, Germany).Centrifuge capable of generating 12,000× *g* acceleration and cooling samples (Eppendorf, 5424R, Hamburg, Germany).Communicating vessels for linear gradient preparation (Bio-Rad, Model 495 Gradient Former, 1654121, Hercules, CA, USA).Cuvette spectrophotometer capable of measuring absorbance at 240, 280, and 412 nm (Agilent, Varian Cary Eclipse Fluorescence Spectrophotometer, Santa Clara, CA, USA).100 mL glass beakers, ×5 (BRAND, BR90624, Wertheim, Germany).250 mL glass beaker (BRAND, BR91236, Wertheim, Germany).500 mL glass beaker (BRAND, BR91217, Wertheim, Germany).2 L glass beakers, ×2 (BRAND, BR91263, Wertheim, Germany).Laboratory balance (Accuris, Precision Balance, Denver, CO, USA).Laboratory ice generator (Fisher Scientific, CurranTaylor™ Scotsman Brand Flake Ice Maker Floor Model, Waltham, MA, USA).Laboratory spatula (Aldrich, Z283274; Scienceware, Z177962, St. Louis, MO, USA).Magnetic stirrer (Fisher Scientific, Thermo Scientific™ Cimarec+™ Stirrer Series, Waltham, MA, USA).pH-meter (Ohaus, Starter 2000, Parsippany, NJ, USA).Peristaltic pump with eluent speed regulation from 1 to 2 mL per min (Shenchen, SP-MiniPump01, Baoding, Hebei, China).Plate reader capable of measuring absorbance at 412, 414 nm (Tecan, Tecan Infinite 200 PRO, Männedorf, Switzerland).Timer (Fisher Scientific, Fisherbrand™ Traceable™ Big-Digit Timer/Stopwatch, Waltham, MA, USA).Tube rotator (ELMI, Intelli-Mixer RM-2S, Riga, Latvia).Vacuum pump (Bio-Rad, Vacuum Station, 1655004, Hercules, CA, USA).

## 3. Procedure

### 3.1. Reagent Preparation

Prepare the required solutions according to Table 1.

### 3.2. H_2_O_2_ Concentration Measurement (10 min)


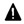
  **CRITICAL STEP.** Establishing the exact concentration of the initial H_2_O_2_ stock is important because the yield of the product in the enzymatic synthesis of HOSCN (Section 3.7) is sensitive to this parameter.

With the use of distilled water, dissolve 8.5 M H_2_O_2_ stock to a concentration with an expected OD_240_ of 0.5 (e.g., mix 1.35 μL of 8.5 M H_2_O_2_ with 998.65 μL of water).Read the absorbance of distilled water at 240 nm with a spectrophotometer.Read the absorbance of the H_2_O_2_ solution at 240 nm with a spectrophotometer.Subtract the background absorbance from the absorbance of the H_2_O_2_ solution.With the use of H_2_O_2_ molar extinction coefficient (43.6 M^−1^ cm^−1^) and the pathlength of light (e.g., 1 cm), calculate the concentration of the analyte.To find the concentration of the stock, multiply the obtained value by the dilution factor (e.g., multiply by 740.74 if diluted as described above).

**Note**: For H_2_O_2_, several molar extinction coefficients have been published in the literature (from 36 to 43.6 M^−1^ cm^−1^ [41,42,43,44]), which may be explained by differences in pH and ionic strength, as well as in equipment accuracy. In this work, we chose a popular value of 43.6 M^−1^ cm^−1^ and recommend using it for this protocol to achieve consistent results.

### 3.3. LPO Purification (1 h 30 min, 8 h 30 min, 3 h 30 min, 6 h)

#### 3.3.1. Removal of Casein and Fat from Unpasteurized Bovine Milk (1 h 30 min)

Defrost 1 L of bovine milk at 25 °C and transfer to a glass beaker cooled by melting ice.**OPTIONAL STEP**. A 1 mL sample of defrosted milk can be collected for the determination of LPO (ABTS-peroxidase assay, Section 3.4) and protein (e.g., Bradford assay) content.Under continuous mixing on a magnetic stirrer (80–100 rpm), add 30 mL of ice-cooled 1 M citric acid to milk. Add 1 mL of 1 M citric acid every 30 s (total time: 15 min).Transfer the obtained suspension to centrifuge tubes and centrifugate at +4–8 °C over 15 min at 2500× *g* in a bucket rotor.After centrifugation, with the use of a thin spatula, remove the upper layer of fat from the walls of the tubes.Separate the yellow-green supernatant of whey from the white sediment and transfer it to a glass beaker on ice.Repeat steps 4–6 with the obtained supernatant one more time.
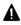
 **CRITICAL STEP**. If the membrane of the capsule filter contains azide as a preservative, it must be extensively washed by 10 mM MES-NaOH buffer (pH 5.5) before whey filtration.To remove the residual traces of fat and casein sediment, filter the collected whey through a capsule membrane filter with a vacuum pump.To collect the residual whey, wash the filter with 55 mL of ice-cooled 10 mM MES-NaOH buffer (pH 5.5), which corresponds to ~10% of the filtrated volume. Total time: 20 min.**OPTIONAL STEP.** A 1 mL sample of filtrated whey can be collected for the determination of LPO (ABTS-peroxidase assay, Section 3.4) and protein (e.g., Bradford assay) content.

#### 3.3.2. Isolation of Cationic Proteins from Whey (8 h 30 min)

Equilibrate the column with a strong cation exchanger resin (12 mL) by 3 volumes (36 mL) of ice-cooled 10 mM MES-NaOH buffer (pH 5.5) with an elution speed of about 2 mL per min. Total time: 18 min.Load the filtrated whey on the equilibrated column with an elution speed of about 2 mL per min. The whey should be kept on ice. Total time: 5 h 10 min.Wash the column with 4 volumes (about 48 mL) of ice-cooled 150 mM NaCl, 10 mM MES-NaOH buffer (pH 5.5), with an elution speed of about 2 mL per min. Total time: 24 min.If the last portions of the eluate are characterized by absorbance at 280 nm higher than 0.03, repeat elution by ice-cooled 150 mM NaCl, 10 mM MES-NaOH buffer (pH 5.5).Start elution by a linear gradient from communicating vessels with 80 mL of ice-cooled 150 mM NaCl, 10 mM MES-NaOH buffer (pH 5.5) and 80 mL of ice-cooled 4 M NaCl, 20 mM phosphate buffer (pH 7.4). The first solution should be continuously mixed by a magnetic stirrer. Keep the speed of elution at about 1 mL per min and collect 8 mL fractions to separate tubes on ice. Total time: 160 min.Combine the fractions with a green-brown color for the next step of LPO purification and store on ice.**OPTIONAL STEP.** The combined fractions can be collected for the determination of LPO (ABTS-peroxidase assay, Section 3.4) and protein (e.g., Bradford assay) content, as well as for Reinheitszahl calculation (Section 3.6).

#### 3.3.3. Separation of LPO by Hydrophobic Chromatography (3 h 30 min)

Add solid (NH_4_)_2_SO_4_ to the ice-cooled combined fractions obtained at the previous step of LPO purification: 1.189 g per 10 mL (final concentration should be 0.9 M). Mix thoroughly until salt dissolution. Total time: 30 min.Centrifugate the obtained suspension at +4–8 °C for 15 min at 2500× *g* in a bucket rotor.Collect the supernatant for LPO separation on a hydrophobic resin and keep on ice.Equilibrate the column with hydrophobic resin (5 mL) by 5 volumes (25 mL) of ice-cooled 900 mM (NH_4_)_2_SO_4_, 20 mM phosphate buffer (pH 7.4), with an elution speed of about 1 mL per min. Total time: 25 min.Load the collected supernatant on the equilibrated column with an elution speed of about 1 mL per min. Total time: 15 min.Wash the column with 5 volumes (about 25 mL) of ice-cooled 900 mM (NH_4_)_2_SO_4_, 20 mM phosphate buffer (pH 7.4), with an elution speed of about 1 mL per min. Total time: 25 min.If the last portions of the eluate are characterized by absorbance at 280 nm higher than 0.03, repeat elution by ice-cooled 900 mM (NH_4_)_2_SO_4_, 20 mM phosphate buffer (pH 7.4).Start elution by a linear gradient from communicating vessels with 45 mL of ice-cooled 900 mM (NH_4_)_2_SO_4_, 20 mM phosphate buffer (pH 7.4) and 45 mL of ice-cooled 20 mM phosphate buffer (pH 7.4). The first solution should be continuously mixed by a magnetic stirrer. Keep the speed of elution at about 1 mL per min and collect 5 mL fractions to separate tubes on ice. Total time: 1 h 30 min.Combine the fractions with a green-brown color for the next step of LPO purification and store on ice.**OPTIONAL STEP**. The combined fractions can be collected for the determination of LPO (ABTS-peroxidase assay, Section 3.4) and protein (e.g., Bradford assay) content, as well as for Reinheitszahl calculation (Section 3.6).

#### 3.3.4. Separation of LPO by Size-Exclusion Chromatography (6 h)

For (NH_4_)_2_SO_4_ removal and final LPO purification, concentrate the combined fractions with MWCO 30 kDa ultrafiltration units until the final volume is 1 mL. Usually, 15 min centrifugation at 2500× *g* in a bucket rotor at +4–8 °C is enough for filtration of 10 mL.After centrifugation, dilute 1 mL of concentrated LPO to 10 mL by ice-cooled 100 mM phosphate buffer (pH 7.4) and keep on ice.Repeat concentration with ultrafiltration units.Equilibrate the column with size-exclusion resin with two volumes of ice-cooled 100 mM phosphate buffer (pH 7.4). Total time: 2 h 30 min.Load the concentrated sample of LPO on the equilibrated column and elute with 2 column volumes of ice-cooled 100 mM phosphate buffer (pH 7.4). Collect 2 mL fractions to separate the tubes and store on ice. Total time: 2 h 30 min.Combine the fractions with a green-brown color.Concentrate them to 1 mL with MWCO 30 kDa ultrafiltration units.**OPTIONAL STEP**. The combined fractions can be collected for the determination of LPO (ABTS-peroxidase assay, Section 3.4) and protein (e.g., Bradford assay) content, as well as for Reinheitszahl calculation (Section 3.6).Divide concentrated LPO to 200–500 μL aliquots in Eppendorf tubes and store at −80 °C.

**Note**: The step-by-step data of a representative LPO purification from 1 L of bovine milk is summarized in Table 2.

### 3.4. LPO Peroxidase Activity Assay with ABTS as a Chromogenic Substrate (15 min)

Prepare 5 mg/L (or 640 nM) LPO standard: Add 1 μL of 100 μM LPO stock solution (1400 U/mg in guaiacol assay) to 1.56 mL of 100 mM MES-NaOH buffer, pH 5.5, in an Eppendorf tube.Transfer 40 μL of 100 mM MES-NaOH buffer, pH 5.5, to all 8 wells of a strip from a 96-well plate (Figure 2).Transfer 20 μL of LPO standard (5 mg/L) or 20 μL of a tested sample to the bottom line (H) of the strip.Mix the solution in line H by pipetting and transfer 20 μL to line F.Repeat mixing and transferring until you reach line B. In the end, 20 μL of the mixture must be removed from line B, and no diluted enzyme or tested samples should be transferred to line A. The latter is used as a negative control line (addition of the chromogenic solution to this well should not cause the development of a green color). Thus, the lines from H to B receive the series of dilutions from 3- to 2187-fold.
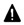
 **CRITICAL STEP.** If tested samples of LPO, obtained after chromatographic fractionation, are visually olive-colored, they must be diluted at least 100 times before transferring to the bottom line (H) of the strip.Repeat steps 1–5 for the tested samples.**OPTIONAL STEP**. If tested samples are characterized by an intense olive color and give overvalues in ABTS-based assay, LPO concentration can be determined by reading absorbance at 412 nm (Section 3.6).Add 160 μL of the chromogenic solution to each well of the strips. Mix thoroughly by pipetting.After 5 min of incubation, read the absorbance of all wells at 414 nm with a plate analyzer.Plot a calibration curve describing the dependence of the sample absorbance on LPO concentration for the standard strip. Calculate concentrations of LPO in the tested samples, taking into account the dilution factors.

### 3.5. LPO Peroxidase Activity Assay with Guaiacol as a Chromogenic Substrate (10 min)

Prepare 2 mg/L LPO in 100 mM sodium phosphate buffer (pH 7.4): Add 1 μL of 100 μM LPO stock solution to 37.8 μL of the buffer. Mix thoroughly. Then, take 2 μL of the resulting solution and mix it with 198 μL of the buffer. Keep on ice.In a 1 mL cuvette, mix 705 μL of 100 mM sodium phosphate buffer (pH 7.4), 250 μL of 100 mM guaiacol solution, and 5 μL of 50 mM H_2_O_2_ solution.Place the cuvette to a spectrophotometer. Set it ready to measure absorbance at 470 nm.Start the reaction by adding 40 μL of 2 mg/L LPO and mix it by pipetting several times.Record absorbance each 2–5 s for a minute.Plot the obtained data in any convenient software (e.g., Microsoft Excel LTSC 2024). Find the linear region of the curve and calculate its slope (ΔOD_470_/Δt).One unit of activity is the amount of enzyme that converts 1 μmol of guaiacol to tetraguaiacol per minute (U = μmol/min). With the use of tetraguaiacol molar extinction coefficient (26,600 M^−1^ cm^−1^) and the pathlength of light (1 cm), calculate the activity of the sample in U/L. Remember that one molecule of tetraguaiacol emerges from the oxidation of four guaiacol molecules.To normalize the obtained activity, divide it by the concentration of LPO in the cuvette (0.08 mg/L).

### 3.6. Reading Absorbance of LPO at 412 nm and Calculating Reinheitszahl (RZ) During Final Steps of LPO Isolation (10 min)

Transfer 100 mM MES-NaOH buffer (pH 5.5) to a cuvette. Next, perform blank absorbance at 280 and 412 nm.Transfer 9 μM pure commercial LPO or a tested LPO sample to a cuvette. Next, measure absorbance at 280 and 412 nm.If the tested samples give overvalues, they should be diluted 4-fold or more to obtain absorbances between 0.7 and 1.6 at both 280 and 412 nm.Calculate the ratio between the absorbance at 412 and 280 nm.If this ratio is more than 0.9, use molar extinction coefficient ε_412_ = 112.3 mM^−1^ cm^−1^ or ε_1% 412_ = 13.9 to determine the LPO concentration. If not, the last step of LPO purification (Section 3.3.4.) should be repeated.

### 3.7. HOSCN Synthesis (20 min)

Choose a buffer system (pH 6.6 or pH 7.4). For more details, go to Section 4 Expected Results.Precool a centrifuge to 5 °C.Prepare 250 μL of 4 μM LPO solution (1400 U/mg in guaiacol assay) in the buffer (e.g., mix 10 μL of 100 μM LPO and 240 μL of the chosen buffer). Place the tube on ice (Figure 3).Add 50 μL of ice-cooled 20 mM NaSCN and 20 mM H_2_O_2_ mixture. Gently mix by pipetting. Avoid bubbling.Incubate for 30 s.Repeat steps 4–5 four more times.To eliminate excess H_2_O_2_, add bovine liver catalase to the final concentration of 100 μg/mL or 200–500 U/mL (e.g., add 5 μL of 10 mg/mL catalase solution to 500 μL of the reaction mixture).If working with pH 6.6, immediately adjust pH to 7.4 via the addition of 4 μL 5 M NaOH in order to decrease HOSCN reactivity.Incubate on ice for 3 min.To obtain the low-molecular-weight fraction of HOSCN and salts, perform ultrafiltration with MWCO 10 kDa columns at 12,000× *g*, 5 °C for 10 min.Store the obtained filtrate on ice.

**Note**: This procedure can be scaled up to larger volumes while maintaining concentrations and ratios.

**Note**: This procedure can be performed in other buffer systems without significant loss in HOSCN yield (in particular, we tested Hanks’ Balanced Salt Solution and Earle’s balanced salt solution). However, it should be kept in mind that the yield is pH-dependent.

### 3.8. HOSCN Concentration Measurement (10 min)

**Note**: This procedure is based on Hawkins et al. [45].

Transfer 100 μL of 100 mM sodium phosphate buffer (pH 7.4) to a single well (#1) of the plate (Figure 4).Transfer 97.5 μL of 200 μM TNB working solution to 6 wells (#2–7) of the plate.Transfer 2.5 μL of 100 mM phosphate buffer (pH 7.4) to wells #2–4.Transfer 2.5 μL of tested HOSCN solution to wells #5–7.Thoroughly mix all solutions with pipetting. Avoid bubbling.Immediately read the absorbance of all wells at 412 nm with a plate analyzer.Subtract the optical density of well #1 from the optical densities of wells #2–7 to correct for background absorbance.Find mean values for wells #2–4 and #5–7.With the use of TNB molar extinction coefficient (14,150 M^−1^ cm^−1^) and the pathlength of light (0.28 cm) calculate the mean concentrations of TNB.Subtract the mean concentration of TNB in wells #5–7 from the mean concentration in wells #2–4. The difference between concentrations represents the amount of TNB consumed in the reaction with HOSCN.To find the concentration of HOSCN in the initial solution, multiply the obtained value by 20. This procedure takes into account both the 40-fold dilution of the sample and the fact that one HOSCN molecule oxidizes two TNB molecules.

**Note**: This procedure can be performed with a standard cuvette spectrophotometer, provided that the volumes are corrected.

### 3.9. Troubleshooting

**Problem**: Low HOSCN yield.

Possible cause 1: High temperature of the reagents.

Solution: All reagents for the synthesis (LPO solution in the buffer, NaSCN and H_2_O_2_ mixture) should be cooled on ice for sufficient time.

Possible cause 2: The pH value of the buffer system is different from the calculated.

Solution: Recalibrate your pH-meter and repeat buffer preparation.

Possible cause 3: The concentration of H_2_O_2_ is different from expected.

Solution: Remeasure the H_2_O_2_ concentration according to Section 3.2.

Possible cause 4: The mixing is performed too vigorously, which leads to bubbling of the mixture and enzyme denaturation.

Solution: Add NaSCN and H_2_O_2_ aliquots smoothly and avoid bubbling during pipetting.

### 3.10. General Notes

There is experimental evidence for a possible non-enzymatic reaction between excess H_2_O_2_ and HOSCN, leading to the formation of byproducts such as cyanosulfurous acid (HOOSCN). HOOSCN is believed to be a more cytotoxic agent with a broader reactivity range than HOSCN [46]. This fact should be taken into account when planning and interpreting experiments modeling oxidative stress in cells.If a pH adjustment procedure is involved during HOSCN synthesis (Section 3.7, Step 8), we advise testing that the amount of NaOH added is sufficient for reaching the target value beforehand. It can be performed with a pH-meter in a large volume and recalculated for the desired volume of the synthesis.

## 4. Expected Results

In the literature, we found 18 sources that describe HOSCN synthesis protocols. We did not include articles dedicated to the investigation of LPO kinetic properties, and focused only on those that provided instructions on the preparative synthesis of HOSCN for practical purposes. On the basis of their structure, we divided these protocols into the eight groups listed below. It is worth noting that individual protocols within groups may differ slightly from each other in the concentrations of components, but the general logic remains the same.

*Group 1* [47]. The synthesis was performed in 368 μL of 10 mM potassium phosphate buffer (pH 7.4) with 67 mM Na_2_SO_4_ at 22 °C. This volume included all components except catalase. The sample contained 0.43 μM LPO (1400 U/mg in guaiacol assay) plus 391 μM NaSCN, and the reaction was started by adding 32 μL of 3.3 mM H_2_O_2_ up to a final concentration of 287 μM. The liquid was thoroughly mixed by pipetting and incubated for 5 min at 22 °C. To eliminate the excess H_2_O_2_, 32 μL of 20 μg/mL bovine liver catalase (Sigma-Aldrich, 2000–5000 U/mg) was used without incubation. Centrifugation to separate protein components of the mixture was not conducted.

*Group 2* [48]. The synthesis was performed in 500 μL of 100 mM sodium phosphate buffer (pH 7.4) at 22 °C. This volume included all components except catalase. The sample contained 0.4 μM LPO (1400 U/mg in guaiacol assay) plus 2 mM NaSCN, and the reaction was started by 5 μL of 25 mM H_2_O_2_. Four more H_2_O_2_ aliquots of the same volume were added to the mixture 1 min apart (up to a final concentration of 1.25 mM). To eliminate the excess H_2_O_2_, bovine liver catalase (Sigma-Aldrich, 2000–5000 U/mg) was used at a final concentration of 40 μg/mL (80–200 U/mL) without incubation. The low-molecular-weight fraction of HOSCN and salts was obtained via filtration with Vivaspin 500 columns (MWCO 10 kDa; Sartorius) at 12,000 g, 10 min, 5 °C.

*Group 3* [4,18,49,50]. The synthesis was performed in 500 μL of 10 mM potassium phosphate buffer (pH 6.6) at 22 °C. This volume included all components except catalase. The sample contained 2 μM LPO (1400 U/mg in guaiacol assay) plus 7.5 mM NaSCN, and the reaction was started by adding 75 μL of 25 mM H_2_O_2_ up to a final concentration of 3.75 mM. The liquid was thoroughly mixed by pipetting and incubated for 15 min at 22 °C. To eliminate the excess H_2_O_2_, bovine liver catalase (Sigma-Aldrich, 2000–5000 U/mg) was used at the final concentration of 40 μg/mL (80–200 U/mL) without incubation. The low-molecular-weight fraction of HOSCN and salts was obtained via filtration with Vivaspin 500 columns (MWCO 10 kDa; Sartorius) at 12,000 g, 10 min, 5 °C.

*Group 4* [51,52,53]. The synthesis was performed in 500 μL of 10 mM potassium phosphate buffer (pH 6.6) at 22 °C. This volume included all components except catalase. The sample contained 2 μM LPO (1400 U/mg in guaiacol assay) plus 7.5 mM NaSCN, and the reaction was started by 5 μL of 75 mM H_2_O_2_. Four more H_2_O_2_ aliquots of the same volume were added to the mixture 1 min apart (up to the final concentration of 3.75 mM), and after the final one, the sample was incubated for 10 min at 22 °C. To eliminate the excess H_2_O_2_, bovine liver catalase (Sigma-Aldrich, 2000–5000 U/mg) was used at a final concentration of 40 μg/mL (80–200 U/mL) without incubation. The low-molecular-weight fraction of HOSCN and salts was obtained via filtration with Vivaspin 500 columns (MWCO 10 kDa; Sartorius) at 12,000 g, 10 min, 5 °C.

*Group 5* [39,40]. The synthesis was performed in 500 μL of 10 mM potassium phosphate buffer (pH 6.6) on ice. This volume included all components except catalase. The sample contained 2 μM LPO (1400 U/mg in guaiacol assay) plus 7.5 mM NaSCN, and the reaction was started by 5 μL of 75 mM H_2_O_2_. Three more H_2_O_2_ aliquots of the same volume were added to the mixture 1 min apart (up to the final concentration of 3 mM). To eliminate the excess H_2_O_2_, bovine liver catalase (Sigma-Aldrich, 2000–5000 U/mg) was used at a final concentration of 10 μg/mL (20–50 U/mL) without incubation. The low-molecular-weight fraction of HOSCN and salts was obtained via filtration with Vivaspin 500 columns (MWCO 10 kDa; Sartorius) at 12,000 g, 10 min, 5 °C.

*Group 6* [54,55,56]. The synthesis was performed in 500 μL of 10 mM potassium phosphate buffer (pH 6.6) at 22 °C. This volume included all components. The sample contained 1.3 μM LPO (1400 U/mg in guaiacol assay) plus 7.5 mM NaSCN, and the reaction was started by 5 μL of 75 mM H_2_O_2_. Three more H_2_O_2_ aliquots of the same volume were added to the mixture 1 min apart (up to the final concentration of 3 mM). Bovine liver catalase was not used. The low-molecular-weight fraction of HOSCN and salts was obtained via filtration with Vivaspin 500 columns (MWCO 10 kDa; Sartorius) at 12,000 g, 10 min, 5 °C.

*Group 7* [6,57,58]. The synthesis was performed in 500 μL of 10 mM potassium phosphate buffer (pH 7.4) at 22 °C. This volume included all components except catalase. The sample contained 1.2 μM LPO (1400 U/mg in guaiacol assay) plus 7.5 mM NaSCN, and the reaction was started by 5 μL of 80 mM H_2_O_2_. Two more H_2_O_2_ aliquots of the same volume were added to the mixture 1 min apart (up to a final concentration of 2.4 mM), and after the final one, the sample was incubated for 10 min on ice. To eliminate the excess H_2_O_2_, bovine liver catalase (Sigma-Aldrich, 2000–5000 U/mg) was used at a final concentration of 30 μg/mL (60–150 U/mL) without incubation. The low-molecular-weight fraction of HOSCN and salts was obtained via filtration with Vivaspin 500 columns (MWCO 10 kDa; Sartorius) at 12,000 g, 10 min, 5 °C.

*Group 8* [5]. The synthesis was performed in 500 μL of phosphate-buffer saline (137 mM NaCl, 2.7 mM KCl, 11.8 mM phosphates, pH 7.4) at 22 °C. This volume included all components except catalase. The sample contained 5.16 μM LPO (1400 U/mg in guaiacol assay) plus 6.5 mM NaSCN, and the reaction was started by 5 μL of 100 mM H_2_O_2_. Four more H_2_O_2_ aliquots of the same volume were added to the mixture 1 min apart (up to a final concentration of 5 mM). To eliminate the excess H_2_O_2_, bovine liver catalase (Sigma-Aldrich, 2000–5000 U/mg) was used at a final concentration of 30 μg/mL (60–150 U/mL) without incubation. The low-molecular-weight fraction of HOSCN and salts was obtained via filtration with Vivaspin 500 columns (MWCO 10 kDa; Sartorius) at 12,000 g, 10 min, 5 °C.

It is possible to highlight several main factors that distinguish these groups, as follows: (1) NaSCN/H_2_O_2_ ratio; (2) the number of H_2_O_2_ additions; (3) the buffer system used; (4) the incubation time before catalase addition; and (5) the temperature of the reaction mix. Table 3 summarizes this information, as well as the HOSCH yields we obtained in each case after following all the guidelines.

Despite the fact that for Group 1, we observed 100% conversion of H_2_O_2_ to HOSCN, the total yield was very low (~0.3 mM) due to the low initial concentrations of the reagents. Therefore, this protocol cannot be used for experiments with high cell density. Unfortunately, it cannot be simply upscaled either, since excess H_2_O_2_ is known to inhibit LPO by converting it into Compound III with subsequent heme cleavage and iron liberation [29]. Moreover, H_2_O_2_ demonstrates reactivity towards HOSCN, the target product, facilitating its degradation to HOCN, HCN, H_2_SO_3_, and H_2_SO_4_ species [59]. We faced this in our practice when trying to scale up our own protocols by simply increasing all concentrations. This idea can be further illustrated by a comparison of Groups 2 and 8, which have a very similar NaSCN/H_2_O_2_ (~1.3–1.6) ratio, the same buffer pH (7.4), temperature (22 °C), and total incubation time (5 min), but the absolute concentrations in Group 8 are about three times larger. However, the observed increase in the absolute yield is 1.5 rather than 3 times.

A comparison of Groups 3 and 4, which differ only in whether the entire bolus of H_2_O_2_ is given at once or split into separate additives, shows that the interactions in this system are even more complicated. It seems likely that the conditions of excess NaSCN over H_2_O_2_ observed in the initial steps of the protocol are not suitable for high enzymatic activity [60], and the benefit from reduced enzyme inhibition by high H_2_O_2_ becomes irrelevant.

The maximum yield (~1.85 mM) was observed for Group 5. Apparently, this was due to the following two facts: a lower pH value (6.6), which is more optimal for LPO catalysis [60], and a lower synthesis temperature, which stabilizes HOSCN and prevents its decomposition [48]. Theoretically, part of the target product is consumed in nonspecific reactions with catalase; however, its addition cannot be completely omitted, since residual H_2_O_2_ will have its own effects on cells.

Given that none of the published protocols satisfied our demands for total HOSCN yield, we decided to perform optimization. The general ideas behind our modifications were as follows. First, the total H_2_O_2_ bolus should be divided to several aliquots. However, it was shown before this, the optimal NaSCN/H_2_O_2_ ratio for HOSCN production is about 1 [60]. We, therefore, suggest adding not H_2_O_2_ itself, but its mixture with NaSCN to maintain their relative proportions throughout the whole synthesis. Second, the sample should be kept on ice in order to decrease HOSCN decomposition and nonspecific reactions with protein components [48]. Third, the intervals between additions should be as small as possible to shorten the whole synthesis time, since a significant part of the product can degrade during long preparation. We also propose a pH switching procedure, namely performing the LPO reaction at 6.6 and then adjusting the pH to 7.4 with NaOH, since HOSCN is stabilized under these conditions [61]. The described modifications (named Protocol 1 in Table 3) allowed for increasing total product yield up to 2.93 mM, which is 60% better than that in published guidelines. We also tested a simplified version of the procedure (Protocol 2), where all steps were performed at pH 7.4. This gave a total product yield of 2.49 mM, which is still 35% better than other analogs. The optimized conditions are described in detail in Section 3.7.

Finally, we measured the stability of HOSCN produced in the described system over three hours (Figure 5). Storage on ice significantly inhibits oxidant degradation; however, even in this case, approximately 5% of the compound is lost within 10 min. Considering that HOSCN concentration measurement requires time (for mixing reagents, incubation, and absorbance reading), certain amendments should be introduced when calculating the doses received by treated cells. Moreover, after 15 min on ice, only 90% of the oxidant remains in the solution. This highlights the need to have reagents and equipment for concentration measurement fully prepared upon the completion of synthesis. The reaction between 2-nitro-5-thiobenzoic acid (TNB) and HOSCN is characterized by a kinetic constant of 4·10^5^ M^−1^ s^−1^ (pH 7.4) [26], so the absorbance of the test solution can be registered immediately without a need for prolonged incubation (Figure 6).

Alkaline hydrolysis of (SCN)_2_ and SCN^−^ oxidation by HOCl are common non-enzymatic alternatives of HOSCN synthesis [26]. Their common limitation is the need to perform the reactions at biologically irrelevant high pH values. Therefore, pH adjustment procedures are strictly required before using the obtained stock on cell cultures, which may be inconvenient in small volumes and, if performed with a pH-meter, take enough time for a significant portion of the oxidant to degrade. The current protocol allows for obtaining HOSCN stock solutions with a concentration of about 2.5 mM in popular buffer systems, like sodium phosphate buffer, phosphate-buffer saline, Hanks’ Balanced Salt Solution, and Earle’s balanced salt solution, all of which have physiological pH values and can be directly added to cells in large volumes without affecting the acidity of the medium. At the same time, the best published enzymatic alternative with pH 7.4 (Group 7; Meredith et al. [6], Nagy et al. [57], Gray [58]) gives solutions with a concentration of about 1.4 mM only. Therefore, the observed improvement in the product yield reaches 79%. Alkaline hydrolysis of (SCN)_2_ also requires preparation of this compound by oxidizing SCN^−^ salts in organic media with Br_2_, which is not convenient in many biology-oriented laboratories [26]. As for SCN^−^ oxidation by HOCl, it has to be performed under the conditions of huge SCN^−^ excess to avoid the generation of overoxidized species, which may have their own effects on cells [26,27]. However, it should be kept in mind that enzymatic HOSCN synthesis has its own limitations. The procedure depends on the quality of the used enzymes, and given the complex kinetics of LPO, which was discussed above, it is not unexpected that even slight deviations in the conditions may significantly affect the reaction yield or the generation of byproducts. Insufficient catalase activity will result in noncomplete H_2_O_2_ elimination, which will compromise experiments with cells, since this oxidant has its own biological functions.

## Figures and Tables

**Figure 1 mps-08-00144-f001:**
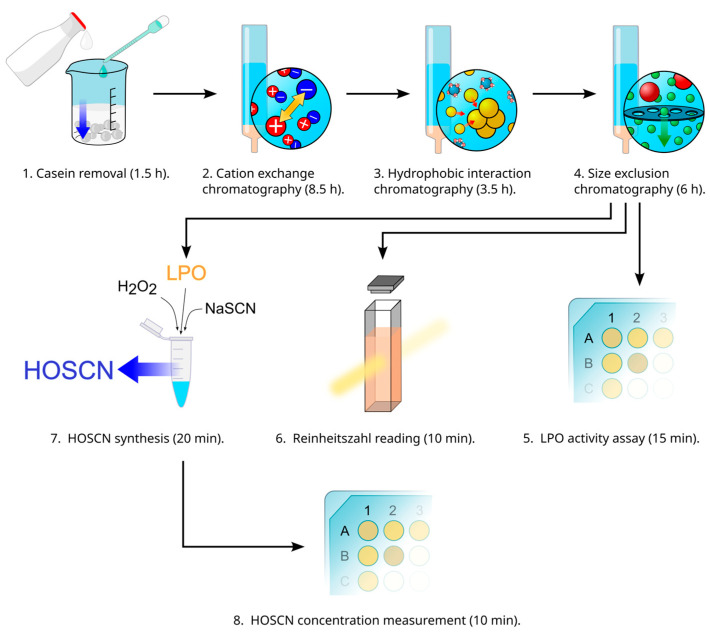
The general scheme of the protocol.

**Figure 2 mps-08-00144-f002:**
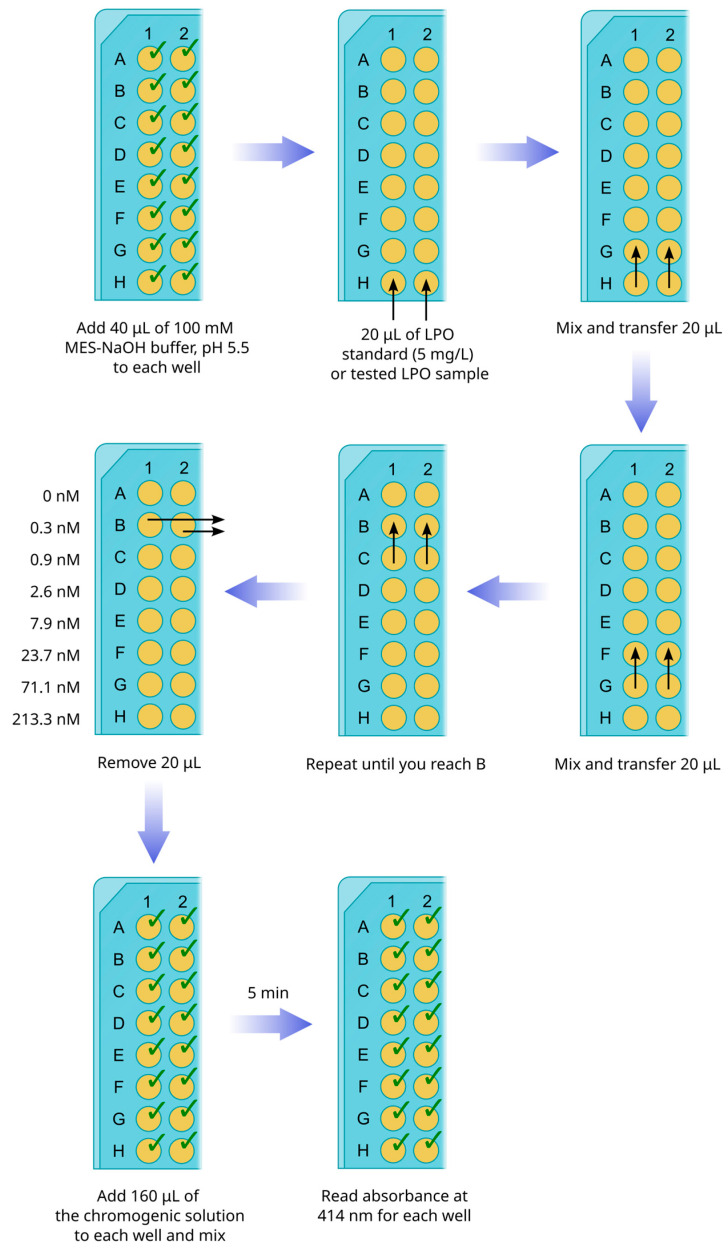
The scheme of plate handling for LPO peroxidase activity assay with ABTS as a chromogenic substrate. The black arrows indicate the direction of liquid transfer.

**Figure 3 mps-08-00144-f003:**
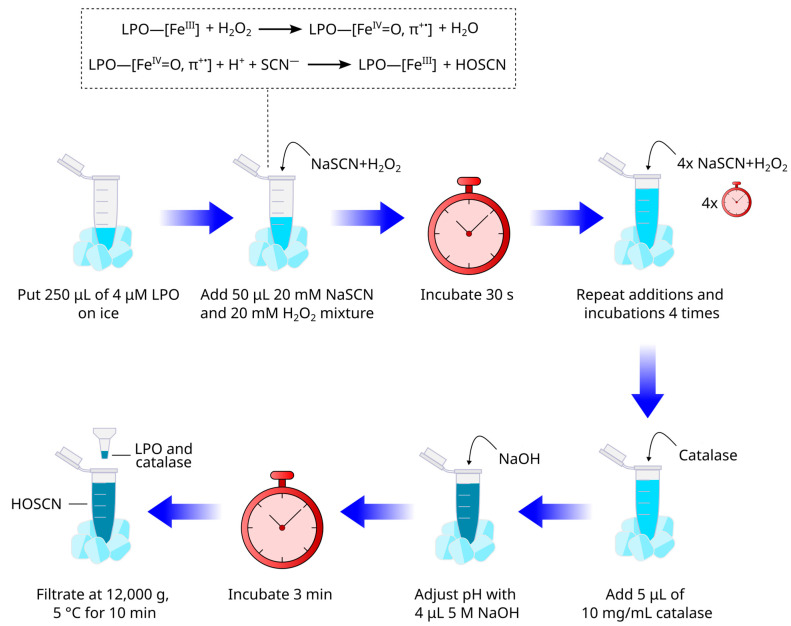
The scheme of HOSCN synthesis with the use of LPO. [Fe^X^] represents the heme and the oxidation state of iron. π^+•^ represents the porphyrin π-cation radical.

**Figure 4 mps-08-00144-f004:**
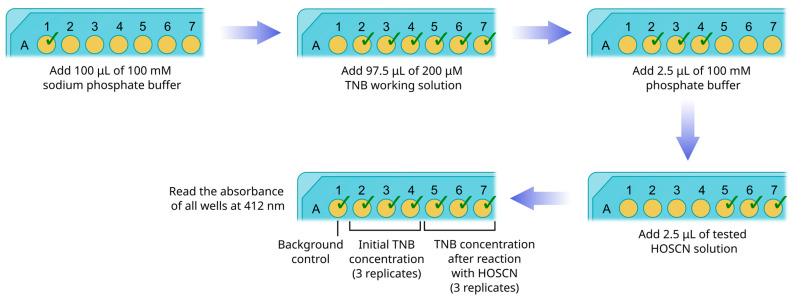
The scheme of plate handling for HOSCN concentration measurement.

**Figure 5 mps-08-00144-f005:**
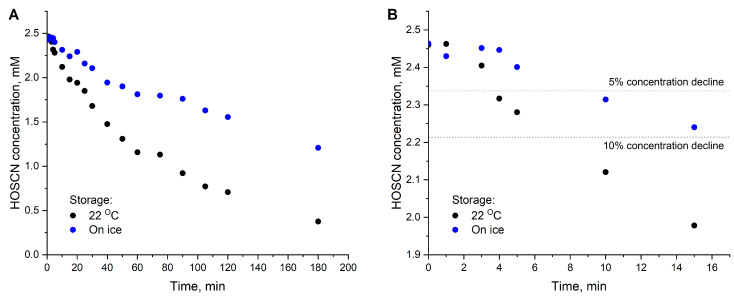
HOSCN stability in 100 mM sodium phosphate buffer, pH 7.4. (**A**) Over 3 h. (**B**) The same data over 15 min. HOSCN was synthesized from H_2_O_2_ and NaSCN as described in Section 3.7. The concentration was measured at different time points during the incubation of the HOSCN stock solution as described in Section 3.8. The initial concentration of the HOSCN stock solution was 2.46 mM. Measurements were performed in single replicates.

**Figure 6 mps-08-00144-f006:**
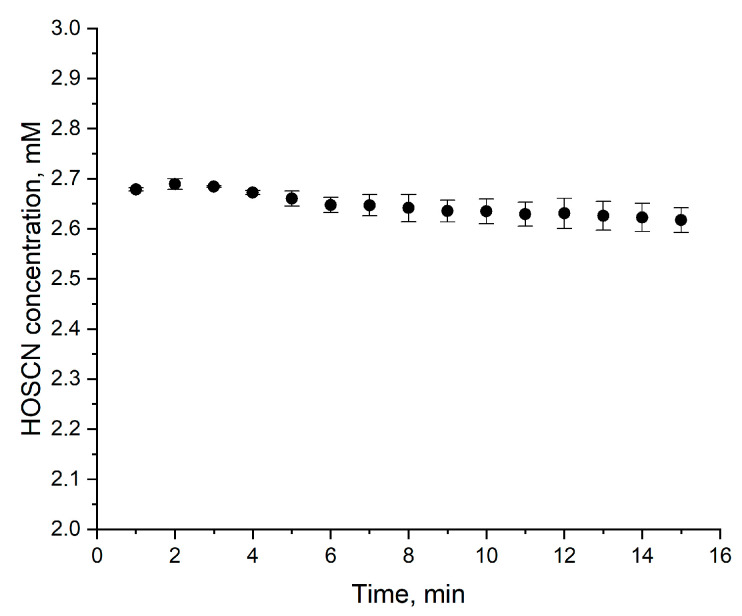
Dependence of HOSCN concentration assay results on plate incubation time. HOSCN was synthesized from H_2_O_2_ and NaSCN as described in Section 3.7. The concentration was measured as described in Section 3.8, except that the plates were not tested immediately but were incubated for various periods of time before absorbance reading. The data are shown as mean ± SD, n = 2.

**Table 1 mps-08-00144-t001:** Detailed recipes for the solutions required for the protocol. n/a—not applicable.

Reagent	Final Concentration	Quantity or Volume
100 mM phosphate buffer, pH 7.4		
Na_2_HPO_4_·7H_2_O	75 mM	1.011 g
NaH_2_PO_4_·H_2_O	25 mM	170 mg
Distilled water	n/a	up to 50 mL
Total	n/a	50 mL
a. Adjust pH using dry phosphates, lab spatula, and pH-meter. b. Store at room temperature.		
100 mM phosphate buffer, pH 6.6		
Na_2_HPO_4_·7H_2_O	40 mM	538 mg
NaH_2_PO_4_·H_2_O	60 mM	413 mg
Distilled water	n/a	up to 50 mL
Total	n/a	50 mL
a. Adjust pH using dry phosphates, lab spatula, and pH-meter. b. Store at room temperature.		
200 mM phosphate buffer, pH 7.4		
Na_2_HPO_4_·7H_2_O	150 mM	4.044 g
NaH_2_PO_4_·H_2_O	50 mM	680 mg
Distilled water	n/a	up to 100 mL
Total	n/a	100 mL
a. Adjust pH using dry phosphates, lab spatula, and pH-meter. b. Store at room temperature.		
20 mM phosphate buffer, pH 7.4		
200 mM phosphate buffer, pH 7.4	n/a	5 mL
Distilled water	n/a	45 mL
Total	n/a	50 mL
a. Store at room temperature.		
4 M NaCl, 20 mM phosphate buffer, pH 7.4		
NaCl	4 M	23.376 g
200 mM phosphate buffer, pH 7.4	n/a	10 mL
Distilled water	n/a	up to 100 mL
Total	n/a	100 mL
a. Store at room temperature.		
900 mM (NH_4_)_2_SO_4_, 20 mM phosphate buffer, pH 7.4		
(NH_4_)_2_SO_4_	900 mM	11.893 g
200 mM phosphate buffer, pH 7.4	n/a	10 mL
Distilled water	n/a	up to 100 mL
Total	n/a	100 mL
a. Store at room temperature.		
100 mM MES-NaOH buffer, pH 5.5		
MES·H_2_O	100 mM	10.66 g
NaOH	20 mM	400 mg
Distilled water	n/a	up to 500 mL
Total	n/a	500 mL
a. Adjust pH using dry MES·H_2_O and NaOH, lab spatula, and pH-meter. b. Store at +4 °C.		
10 mM MES-NaOH buffer, pH 5.5		
100 mM MES-NaOH buffer, pH 5.5	n/a	10 mL
Distilled water	n/a	90 mL
Total	n/a	100 mL
a. Store at +4 °C.		
150 mM NaCl, 10 mM MES-NaOH buffer, pH 5.5		
NaCl	150 mM	1.753 g
100 mM MES-NaOH buffer, pH 5.5	n/a	20 mL
Distilled water	n/a	up to 200 mL
Total	n/a	200 mL
a. Store at +4 °C.		
1 M citric acid solution		
C_6_H_8_O_7_·H_2_O	1 M	21 g
Distilled water	n/a	up to 100 mL
Total	n/a	100 mL
a. Store at +4 °C.		
10 mM ABTS stock solution		
ABTS	10 mM	55 mg
Distilled water	n/a	up to 10 mL
Total	n/a	10 mL
a. Store at +4 °C.		
10 mM H_2_O_2_ stock solution		
8.5 M H_2_O_2_	10 mM	10 μL
Distilled water	n/a	8.490 mL
Total	n/a	8.5 mL
a. Store at +4 °C.		
Chromogenic solution		
10 mM ABTS stock solution	150 mM	1.8 mL
10 mM H_2_O_2_ stock solution	n/a	0.9 mL
100 mM MES-NaOH buffer, pH 5.5	n/a	15.3 mL
Total	n/a	18 mL
a. Chromogenic solution can be stored in a dark place at room temperature for no more than 2 h.		
50 mM H_2_O_2_ stock solution		
8.5 M H_2_O_2_	50 mM	5.9 μL
100 mM phosphate buffer, pH 7.4	n/a	994.1 μL
Total	n/a	1000 μL
a. Prepare right before the LPO guaiacol assay and do not store.		
100 mM guaiacol stock solution		
9.09 M guaiacol	100 mM	11 uL
100 mM phosphate buffer, pH 7.4	n/a	989 uL
Total	n/a	1000 uL
a. Guaiacol solution can be stored in a dark place at room temperature for no more than one week.		
50 mM NaOH solution		
NaOH	50 mM	0.1 g
Distilled water	n/a	up to 50 mL
Total	n/a	50 mL
a. Store at room temperature.		
8 mM TNB stock solution		
DTNB	5 mM	2 mg
50 mM NaOH solution	n/a	1 mL
Total	n/a	1 mL
a. Place the tube with solution to a rotator and mix on low speed for 30 min. In these conditions, alkaline hydrolysis of DTNB occurs and the expected concentration of 2-nitro-5-thiobenzoic acid (TNB) is about 8 mM. Alternatively, you can vortex the tube for several minutes; however, the degree of hydrolysis will be lower. b. 8 mM TNB stock solution can be stored in a dark place at room temperature for a week without significant decline in the concentration.		
200 μM TNB working solution		
8 mM TNB stock solution	n/a	25 μL
100 mM phosphate buffer, pH 7.4	n/a	975 μL
Total	n/a	1 mL
a. Prepare right before HOSCN concentration assay. b. 1 mL of 200 µM TNB working solution is sufficient for 9 wells in HOSCN concentration assay. c. 200 μM TNB working solution can be stored in a dark place at room temperature for no more than 3 h.		
5 M NaOH solution		
NaOH	5 M	0.1 g
Distilled water	n/a	500 μL
Total	n/a	500 μL
a. Prepare right before HOSCN synthesis and do not store.		
200 mM H_2_O_2_ solution		
8.5 M H_2_O_2_	200 mM	4 μL
100 mM phosphate buffer, pH 7.4 or 6.6	n/a	166 μL
Total	n/a	170 μL
a. Prepare right before HOSCN synthesis and do not store.		
200 mM NaSCN solution		
NaSCN	200 mM	32.8 mg
100 mM phosphate buffer, pH 7.4 or 6.6	n/a	2 mL
Total	n/a	2 mL
a. Prepare right before HOSCN synthesis and do not store.		
20 mM H_2_O_2_ and 20 mM NaSCN solution		
200 mM H_2_O_2_ solution	n/a	100 μL
200 mM NaSCN solution	n/a	100 μL
100 mM phosphate buffer, pH 7.4 or 6.6	n/a	800 μL
Total	n/a	1 mL
a. Prepare right before HOSCN synthesis and do not store.		
10 mg/mL bovine liver catalase solution		
Bovine liver catalase (2000–5000 U/mg)	10 mg/mL	1 mg
100 mM phosphate buffer, pH 7.4	n/a	100 μL
Total	n/a	100 μL
a. Can be divided into 5 µL aliquots and stored at −20 °C for up to one month.		

**Table 2 mps-08-00144-t002:** Step-by-step purification of LPO during isolation from unpasteurized bovine milk. Total protein and active LPO are calculated by multiplying the protein or active LPO concentration by the fraction volume at each step. Fold of purification is calculated as the ratio between active LPO and total protein at this step to the ratio at the first step (29/34230). Yield is calculated as the percentage of initial LPO remaining at each step.

Step	Total Volume, mL	Total Protein, mg	Active LPO, mg	Purification, Fold	Reinheitszahl (A_412_/A_280_)	Yield, %
Defrosted milk	1000	34,230	29	1	-	100
Filtrate after fat and casein removal	615	5320	28.7	6.4	-	99
Eluate from cation-exchange chromatography (UNOSphere S)	16	248	25.2	120	0.069	86.9
Eluate from hydrophobic chromatography (Butyl-Sepharose FF)	10	44.8	13.4	353	0.382	46.2
Ultrafiltrate of eluate	1	23.5	12.4	623	0.409	43.8
Ultrafiltrate after size-exclusion chromatography (Superdex 200)	1	11.5	11.5	1180	0.9	39.7

**Table 3 mps-08-00144-t003:** Brief description of HOSCN synthesis protocols tested and the obtained yields of the product. KP—potassium phosphate, NaP—sodium phosphate, PBS—phosphate-buffer saline; n/a—not applicable. n = 3–4. The product yields are highlighted in bold for greater visibility. The detailed information on each protocol is available in the text.

Group	1	2	3	4	5	6	7	8	Protocol 1	Protocol 2
Parameter	Protocols published in the literature								Protocols described here	
LPO (1400 U/mg in guaiacol assay), μM	0.43	0.4	2	2	2	1.3	1.2	5.16	2	2
H_2_O_2_, mM	0.287	1.25	3.75	3.75	3	3	2.4	5	10	10
NaSCN, mM	0.391	2	7.5	7.5	7.5	7.5	7.5	6.5	10	10
NaSCN/H_2_O_2_	1.36	1.6	2	2	2.5	2.5	3.125	1.3	1	1
Additions, n	1	5	1	5	4	4	3	5	5	5
Time between additions, min	n/a	1	n/a	1	1	1	1	1	0.5	0.5
Buffer	10 mM KP, 67 mM Na_2_SO_4_	100 mM NaP	10 mM KP	10 mM KP	10 mM KP	10 mM KP	10 mM KP	PBS	100 mM NaP	100 mM NaP
pH	7.4	7.4	6.6	6.6	6.6	6.6	7.4	7.4	6.6	7.4
Time before catalase, min	5	1	15	10	1	n/a	10	1	3	3
Catalase (2000–5000 U/mg), μg/mL	1.6	40	40	40	10	n/a	30	30	100	100
Temp, °C	22	22	22	22	ice	22	22	22	ice	ice
Mean yield, mM	**0.29**	**0.79**	**1.52**	**1.12**	**1.86**	**1.60**	**1.38**	**1.17**	**2.93**	**2.49**
Standard deviation, mM	**0.01**	**0.04**	**0.03**	**0.11**	**0.13**	**0.06**	**0.08**	**0.03**	**0.08**	**0.08**

## Data Availability

The raw data supporting the conclusions of this article will be made available by the authors on request.

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
