# Peer review of "An Optimized Protocol for Enzymatic Hypothiocyanous Acid Synthesis"

_mps, 2025, doi:10.3390/mps8060144_

Round 1
Reviewer 1 Report
Comments and Suggestions for Authors
Lines 39/40, the authors state “For instance, reaction between H2O2 and free thiols is characterized by a bimolecular constant of 10-30 M–1 s–1[2]”, yet in said reference is stated that: “For hydrogen peroxide, the calculated rate constants for the reaction with the thiolate anion all fell within the range 18 –26 M–1 s–1”. Although a minor detail, please check
Also, the authors refer to the ratio kcat/KM as a bimolecular rate constant. The ratio kcat/KM is typically designated as a second-order rate constant (observed under certain circumstances) see e.g. https://doi.org/10.1021/acs.jchemed.1c01268, that acts as a bimolecular rate constant and can be used to quantify enzyme efficiency.
Lines 67/68, “HOCl has been widely studied in this context, but the obtained results are often contradictive” should be “HOCl has been widely studied in this context, but the obtained results are often contradictory”
Lines 94/98: “Since it is difficult to control for the physiological effects of overoxidized species, enzymatic HOSCN synthesis with lactoperoxidase (LPO) remains the most popular technique. Multiple articles include corresponding protocols, which usually differ by several factors like SCN–/H2O2 ratio, buffer pH value, the number of substrate additions, or incubation time. In practice, it is almost impossible to rationally choose one of them, because the authors usually do not report the mean yields. Only several papers provide rough estimations” This information highlights some limitation for the use of LPO for the intended goal. May one assume that the complex kinetics of the reaction and the inhibitory role of hydrogen peroxide (see e.g.
https://doi.org/10.3390/antiox10111646) contribute for the challenges faced when this enzymatic synthesis is considered?
Lines 111/112: “For isolation of LPO (pI about 9.6) from bovine milk” Could the authors also add information on the typical molecular weight of bovine milk LPO? It can help further understanding the chromatographic strategies for LPO purification
Lines 130/131: “which is 60% yield improvement compared to the best recipes from the literature” Please specify, with proper references, which are said best recipes from the litereature
Line 235: “With the use of H2O2 molar extinction coefficient (43.6 M–1 cm–1)…” is the value of the molar extinction coefficient aligned with the pH and ionic strength of the solution?
Could the authors add a figure with more explicit details on 96 well microtiter plate handling in 3.4 and 3.7?
“Isolation of cationic proteins from whey (10 h)” Steo 1 is suggested to require 25 minutes with 3 volumes of eluate at 2 mL/min, while in step 3, although 4 volumes of eluate are used, only 20 minutes are suggested, although the flowrate is the same, please check.
Reviewer 2 Report
Comments and Suggestions for Authors
The article describes an optimised protocol for HOSCN preparation, which includes all steps from lactoperoxidase purification to HOSCN concentration measurement. The main advantage of the current protocol is a significant increase in product yield by 60% compared to published alternatives. The article represents comprehensive research and presents a very detailed protocol and will be widely used in research on oxidative stress and the innate immune system's function, as well as the exploration of HOSCN as a potential therapeutic agent. I recommend publication of the article after minor revision; the points for revision are mentioned below.
Introduction
51-53 In living organisms, (pseudo)hypohalous acids are produced in a mixture due to broad substrate specificity of the corresponding enzymes [8], which is further complicated by the fact that blood concentrations of SCN– can vary significantly depending on lifestyle [9].
Is something known about this mixture? Ratio of components, effects of various conditions of the mixture composition? If so, this information should be added here.
Experimental design
383 If the tested samples give overvalues, they must be diluted 4 or more times for the absorbance to be no more than 0.7-1.6 at 280 and 412 nm.
It is not quite a clear sentence. Maybe it should be rephrased as 0.7 at 280 nm and more than 1.6 at 412 nm?
386 If this ratio is more than 0.9, use molar extinction coefficient ε412 = 112.3 mM–1 cm–1 or ε1% 412 = 13.9 to determine LPO concentration
Obviously, here the second variant, where the ratio is lower than 0.9, should at least be mentioned.
Reviewer 3 Report
Comments and Suggestions for Authors
This work reported a protocol for HOSCN preparation. This topic is within the scope of this journal. However, its writing is too poor. The quality of this manuscript need be further improved. Additional data and Figures are required to be provided. Major revision is suggested.
Other comments:
- In "Title" and "Abstract", the full name of HOSCN is required to provide.
- In "Abstract", it lacks the enough data. And the novelty of this work need be highlighted.
- The information of the used enzymes (e.g., catalase, LPO) need be described in detail. Their source and initial activity are needed to give in the text.
- NMR and Mass Spectrometry analysis of HOSCN are required to provide in this work.
- The captions of Figure 2 and Figure 3 are not clear. More performance conditions are required to be provided.
- In this resection "4. Expected Results", the optimized conditions for enzymatic HOSCN synthesis need be well summarized.
- In Table 3, the volume of LPO and catalase is mentioned. Their enzymatic activity need be provided in this Table 3.
- The reaction scheme about the enzymatic HOSCN synthesis should be provided in the text.
- The advantage and disadvantage of this developed method need be well summarized and discussed in the text. The novelty of this work should be well highlighted.
Round 2
Reviewer 3 Report
Comments and Suggestions for Authors
This revised version can be accepted as it is.